# Solve Smart, Not Often: Policy Learning for Costly MILP Re-solving

## Abstract

A common challenge in real-time operations is deciding whether to re-solve an optimization problem or continue using an existing solution. While modern data platforms may collect information at high frequencies, many real-time operations require repeatedly solving computationally intensive optimization problems formulated as Mixed-Integer Linear Programs (MILPs). Determining when to re-solve is, therefore, an economically important question. This problem poses several challenges: 1) How to characterize solution optimality and solving cost; 2) How to detect environmental changes and select beneficial samples for solving the MILP; 3) Given the large time horizon and non-MDP structure, vanilla reinforcement learning (RL) methods are not directly applicable and tend to suffer from value function explosion. Existing literature largely focuses on heuristics, low-data settings, and smooth objectives, with little focus on common NP-hard MILPs. We propose a framework called Proximal Policy Optimization with Change Point Detection (POC), which systematically offers a solution for balancing performance and cost when deciding appropriate re-solving times. Theoretically, we establish the relationship between the number of re-solves and the re-solving cost. To test our framework, we assemble eight synthetic and real-world datasets, and show that POC consistently outperforms existing baselines by 2%-17%. As a side benefit, our work fills the gap in the literature by introducing real-time MILP benchmarks and evaluation criteria.

## 1 Introduction

Combinatorial optimization problems are prevalent in industries such as transportation, energy, and supply chain (Williams, 2013). Since finding the optimal solution is NP-hard, solving large-scale MILPs requires substantial resources and significant computation time. However, in many real-world settings, we encounter high-frequency observational data involving systems that *change dynamically* over time, and frequent re-solving can incur unacceptable costs: either computational costs of excess usage of compute resources, or operational costs associated with the overhead of too frequent changes to the underlying decision variables (e.g., corresponding to a supply chain plan). Most previous studies have focused on orthogonal approaches for reducing computational costs, such as using the previous solution for warm start (Marcucci & Tedrake, 2020; Zhang et al.), whereas this work proposes a more general framework that directly tackles the issue of excess re-solves. In particular, once a re-solve is carried out, our proposed framework can be used in conjunction with the techniques explored in prior studies.

Our problem setting is ubiquitous in real life and carries significant practical importance. For example, in transportation scheduling (Zhang et al., 2020b), Google Maps receives high-frequency streams of traffic data and user search requests. If the shortest path were recalculated based on the latest traffic information for every single user request, it would incur substantial computational costs and cause noticeable search delays. However, when the system undergoes significant changes, such as a traffic accident occurring on certain roads, re-solving for the optimal route seems necessary. Another example is production planning (Cedillo-Robles et al., 2020; Dunke & Nickel, 2023). In the GPU market, demand is constantly shifting, and Nvidia may reallocate production capacity from consumer-grade GPUs, such as the RTX 5090, to data center GPUs, such as the H100. Changes in production lines incur switching costs, while the company's revenue also depends on the evolving market. Therefore, as mentioned above, we consider a general notion of re-solving cost, which

not only covers the computational cost of solving large-scale MILPs, but also accounts for consumer losses caused by delays, as well as the switching cost of transitioning from the old solution to the new one, thereby providing a general re-solving scheduling framework for agents with different needs.

Besides evolving optimizations, another key factor shaping the re-solving policy relates to *uncertainty* about the future (Quionero-Candela et al., 2009), as the exact optimization problem may be typically unobserved and must be estimated from past data. When the environment shifts, outdated samples produce poor estimates, and hence additional new data is required for accurate estimates. An effective policy must therefore trade off the cost of re-solving against the need for reliable estimates. This challenge cannot be handled by standard dynamic programming, as the current policy both depends on past solutions and influences future re-solving decisions. To address this, we propose a policy learning framework with change-point detection that explicitly balances re-solving costs against the need for new estimation samples under environmental shifts.

Our policy, as visualized in Figure 1, highlights two main factors that determine when re-solving is needed. First, if the environment changes significantly, such as shifts in operations costs reflected in the MILP objective, the underlying optimization problem also changes, making re-solving necessary. An effective policy here is to detect major changes in environment dynamics and re-solve when they occur. Second, even in a stable environment, we still need to estimate optimization parameters obtained from sample observations. While more sampled data improves accuracy, the benefit of frequent re-solving decreases over time due to diminishing returns from collecting similar observations. In this case, an effective policy is to gradually reduce the re-solving frequency. We formally establish these characteristics of optimal re-solving schedules and corroborate the effectiveness of the resulting learned policy through extensive experiments, where our approach significantly outperforms existing heuristic methods.

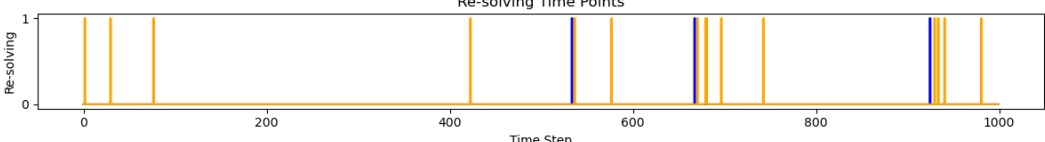

Figure 1: Re-solving decision for an instance: The blue vertical lines represent change points, and the orange lines represent POC re-solving times. We observe that between two change points, the POC re-solving intervals gradually increase.

**Our Contributions.** In this paper, we first study the problem of learning an optimal re-solving policy for varying MILPs. We summarize our main contributions as follows.

First, we introduce a principled decision framework for when to re-solve large-scale, real-time MILPs. We theoretically establish that the number of re-solves is upper-bounded by a function of the re-solving cost and derive structural properties of the re-solving frequency.

Second, we delineate the relationship and key differences between our formulation and vanilla reinforcement learning, and clarify why existing methods are ill-suited for this setting. We propose the POC framework, reducing the cumulative loss over existing baselines by 2%-17%. It strikes a favorable balance between re-solving cost and optimization loss, thereby potentially decreasing the number of re-solves in real-world deployments.

Third, we provide a new benchmark and evaluation protocol for deciding when to re-solve dynamic MILP problems. We curate eight synthetic and real-world datasets across eight MILP families, e.g., set packing and travelling salesman problem, filling the gap in the literature caused by the scarcity of real-time optimization datasets, and offer unified evaluation metrics to facilitate future research.

## 1.1 RELATED WORKS

Our work is closely related to the literature on *change point detection* (Aminikhanghahi & Cook, 2017; Londschien et al., 2023; Li et al., 2024a; Dmitrienko et al., 2022; Dong et al., 2024; Xu et al., 2025; Van den Burg & Williams, 2020; Bifet & Gavalda, 2007; Raab et al., 2020), *re-solving scheduling* (Meignan, 2014; Schieber et al., 2018; Yuan et al., 2022; Zych, 2012; Mannino & Sartor, 2020; Mahadevan & Mathioudakis, 2023; Hoffman et al., 2024; Florence et al., 2025) and *reinforce-*

*ment learning for varying environments* (Padakandla et al., 2020; Mao et al., 2020; Padakandla, 2021; Dmitrienko et al., 2022; Cheung et al., 2023; Baumann et al., 2018; Zhang & Dietterich, 1995; Mao et al., 2016). Due to the space limit, we discuss these and more related works thoroughly in Appendix A.1.

## 2 PRELIMINARIES

We outline our formulation of the re-solving problem. We have an online data flow with an MILP problem. At each time step $t \in [T] = \{1, ..., T\}$, we use $\min_x c_t^T x$ subject to $Ax \leq b$ and $x \in \mathbb{Z}_{\geq 0}$ to represent the problem without loss of generality, and we defer further discussion of different forms of MILPs to Appendix D.1. In practice, constraints often change much more slowly than the objective. For example, while traffic conditions in New York can change rapidly, the road network itself remains fixed; similarly, product prices tend to fluctuate much faster than factory productivity. Mathematically, the change of constraints may lead to an infeasible old solution and make re-solving unavoidable. The objective $c_t$ comes from the distribution $p_t$, and $p_t$ might change over time. We assume $p_t = p_{t-1}$ with probability $1 - \rho_t$ and change arbitrarily with probability $\rho_t$. For example, a traffic accident happens with a certain probability and may alter traffic conditions. Typical systems are stable, so $\rho_t$ is small. People usually assume $\rho_t \lesssim \mathcal{O}(T^{-\kappa})$ (Wei & Srivatsva, 2018) or $\sum_{t=1}^T \rho_t \lesssim o(T)$ (Besson et al., 2022). We access some offline data, denoted by $\{g_i = (A_i, b_i, \{c_{it}\}_{t=1}^{T_i})\}_{i \in [I]}$. We aim to use them to learn a policy for determining appropriate re-solving times in the online setting.

We use $\xi \in \{0,1\}^T$ to denote the re-solving action, where $\xi_t = 1$ if we re-solve the MILP and 0 otherwise. We use $t_k$ to denote the time of the $k$-th re-solving. Besides deciding when to re-solve, we also need to select informative data to predict the evolving environment. Objectives from other distributions offer no benefit for estimating the objective at hand. We assume that $c_t$ is unobservable at time $t$, for instance, a mapping service choosing a route does not know in advance whether an accident will occur en route. Note that this setting is more challenging; however, our framework readily extends to the case where $c_t$ can be observed before the decision is made. Therefore, our action at time t is $\xi_t = \pi^\xi(\mathcal{I}_t)$ and $x_t = \pi^x(\mathcal{I}_t)$ where $\mathcal{I}_t = (A, b, c_1, ..., c_{t-1}, \{g_i\}_{i \in [I]})$ is all available information. We sometimes ignore $A$ and $b$ when it's clear from the context. Then, we can define the cumulative loss

$$CL(\pi) = \underbrace{\sum_{t=1}^T (c_t x_t - c_t x_t^*)}_{\text{Optimization Loss}} + \underbrace{C\|\xi\|_1}_{\text{Re-solving Cost}} \quad ,$$

where $x_t^*$ is the clairvoyant optimal solution and $C$ is the cost of a single re-solving. The re-solving cost may stem from various sources. It could represent the time and computational resources required to obtain a solution, or the monetary cost of invoking an API. It may also encompass other types of costs. For instance, when a mapping service re-computes an optimal route, it may increase latency, reduce user satisfaction, and lead to customer attrition; in supply chain management, changing a hub may incur fixed logistical costs. In the offline phase, we usually have more resources and do not suffer the costs of simulations that have not actually occurred. We assume that the re-solving cost remains constant in the online setting. If it varies, one can simply incorporate it into the state and use the same pipeline. Hence, our goal is to learn $\pi^* = \underset{\pi=(\pi^\xi, \pi^x)}{\arg\min} CL(\pi)$. We illustrate the pipeline in Figure 2.

## 3 THEORETICAL ANALYSIS

Unlike traditional RL, the objective function, represented by $c_t$ is not a Markovian decision process (MDP). At time t, if we simply use $c_{t-1}$, the latest objective we can observe as the state, since we need to select previous data to estimate the environment, using $c_1, ..., c_{t-1}$ will lead to a different transition kernel and reward compared with only using $c_{t-1}$. For instance, note that estimating $c_t$ by $\mathbb{E}_{p_t} c_t$ will yield the best online solution due to the linear structure. As we need to estimate the distribution of the upcoming $c_t$, leveraging more previous observations will render a smaller variance. Another option is to use $(c_1, ..., c_{t-1})$ as the state, which covers all available information.

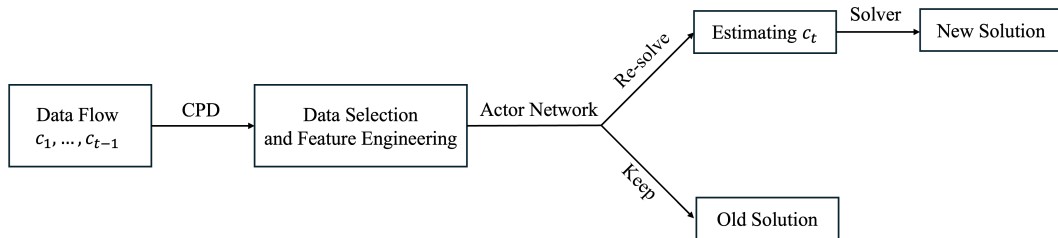

Figure 2: The POC pipeline: We first use the change point detector to choose informative data and construct features. Then, we make re-solving decisions. If we re-solve, we use informative data to estimate and solve.

However, it results in an increasing state dimension and time-varying transition, and we still need to tailor RL algorithms to bypass these issues. In practice, we can use some feature extraction methods to construct the state. Theoretically, we make the following assumptions.

**Assumption 1.** *We assume $\rho_t$, the probability of environment change, is homogeneous with respect to $t$, that is, $\rho_t = \rho$ for all $t$. When $p_t \neq p_{t-1}$, as the environment can change arbitrarily, all previous solutions have approximately the same optimization loss.*

**Assumption 2.** *There exists a feature extractor $\phi$ such that the state representation $s_t = \phi(c_1, ..., c_{t-1})$ fully summarizes all available estimates of the environment at time $t$.*

Assumption 1 implies that if the environment changes, the expected degradation of previous solutions is homogeneous. For example, a traffic accident on the road is usually unrelated to the route we previously selected. Assumption 2 is a standard one in RL with function approximation (Sutton et al., 1999; Prashanth & Bhatnagar, 2010; Jin et al., 2020). Interested readers may refer to Uehara et al. (2021) for details on how to learn the extractor $\phi$. It's realistic in our setup. For instance, if the age of the latest solution is used as a feature, choosing to re-solve at time $t$ resets it to 0, while choosing not to re-solve increases it by 1, independent of previous features. With $s_t$ in hands, we can define optimal value function $V^*(s_t) = \max \mathbb{E}[\sum_{\tau=t}^{T}(c_\tau x_\tau^* - c_\tau x_\tau - C\xi_\tau)]$ and corresponding optimal policy $\pi^*$. Note that we maximize the value function to be consistent with the literature. We then have the following theorem.

**Theorem 1.** *Under Assumptions 1 and 2, learning $\pi^*$ with value function $V^*$ is equivalent to learning $\pi^*$ with a discount value function $\widetilde{V}^*$ with discount rate $1-\rho$ and reward $(1-\rho)(c_t x_t^* - c_t x_t) - C\xi_t$.*

Theorem 1 establishes that, in our setting, environmental changes are theoretically equivalent to having a discount factor strictly smaller than 1. In practice, when $\rho$ is unknown, one may either adaptively estimate $\rho$ from data or employ a fixed discount factor to mitigate the variance of the algorithm.

We now analyze the properties of the number of re-solvings within a stable environment. Recall that when the dynamic doesn't change, we still need to re-solve, as with more observations of the environment, our estimation would be more accurate. For instance, due to the concentration analysis (Wainwright, 2019), we know that the estimation error of $\mathbb{E}[c_t]$ will decay with rate $\mathcal{O}(\frac{1}{\sqrt{n}})$ with $n$ observations. We now make the following assumption on the optimization loss.

**Assumption 3.** *Within a stable environment, if there are $n$ observations to estimate $c_t$, it holds that the expected optimization loss $\mathbb{E}[c_t^T x_t - c_t^T x_t^*]$ of this solution is $\Theta(n^{-\alpha})$. To be specific, we assume there exists $L$ and $U$ such that $\frac{L}{1-\rho}n^{-\alpha} \leq \mathbb{E}[c_t^T x_t - c_t^T x_t^*] \leq \frac{U}{1-\rho}n^{-\alpha}$.*

Assumption 3 is common in the literature, and we discuss how to relax it in Appendix B.5. El Balghiti et al. (2019) proves an aligned upper and lower bound of order $\alpha = \frac{1}{2}$, while Liu & Grigas (2021) provides an upper bound of order $\alpha = \frac{1}{4}$ under loser assumptions. Hu et al. (2022), on the other hand, gives lower bounds of order $\alpha = \frac{1}{2}$ or $\alpha = \frac{1+a}{2+a}$ for some $a$ separately for independent and dependent noise. In our setting, it's also easy to satisfy this assumption. When the optimal solution $x_t^*$ is bounded, we will have an upper bound with $\alpha$ at least $\frac{1}{2}$. Please refer to Example 1 in Appendix B.2.

Unlike Assumption 1, which characterizes the optimization loss arising from different distributions, Assumption 3 characterizes the properties of the optimization loss within the same distribution. Recall that the information at time 1 is an empty set with respect to $\{c_1, ..., c_T\}$, so we can only use a default solution as there is no observation of the objective. We use $t_0 = 1$ to represent the default solution at time 1. We now give some properties of the re-solving times within a period. Here, at time $t_k$, we have $t_k - 1$ samples to estimate $c_t$'s distribution.

**Theorem 2.** *Assuming conditional on the event that there is no change of the environment, under Assumptions 1 to 3, for the optimal strategy, the gap between $t_k^*$ and $t_{k+1}^*$ is at least* $[\frac{L}{U(t_k^*-1)^{-\alpha}-\rho C}]^{1/\alpha} - (t_k^* - 1)$. *In other words, it holds that*

$$t_{k+1}^* \geq 1 + [\frac{L}{U(t_k^* - 1)^{-\alpha} - \rho C}]^{1/\alpha}.$$

Note that $t_{k+1}^*$ is increasing with respect to $t_k^*$, so we can give the lower bound of every $t_k^*$ denoted by $\underline{t_k}$. Since we have a default solution at time 1 and $t_1^*$ should be an integer, from Theorem 2, it holds that $t_1^* \geq 2 := \underline{t_1}$. We then have the following corollaries.

**Corollary 1.** *It holds that every $t_k$ has a lower bound satisfying $\underline{t_0} = 1$, $\underline{t_1} = 2$, and the recurrence*

$$\underline{t_{k+1}} = 1 + [\frac{L}{U(\underline{t_k} - 1)^{-\alpha} - \rho C}]^{1/\alpha}.$$

**Corollary 2.** *When the re-solving cost is large, say $C \geq \frac{U}{\rho}$, we only need to re-solve once at the beginning.*

We derive Corollary 2 as the denominator becomes negative when re-solving cost $C$ is large, which means that compared with the optimization loss incurred by using the old solution, the re-solving cost is never worthwhile. It also demonstrates the relationship between the environment change probability $\rho$ and the re-solving times. When the changing probability $\rho$ is small, re-solving has more benefits for the future, so we intend to re-solve more times.

We now characterize the properties of the intervals between $\underline{t_k}$ and have the following theorem.

**Theorem 3.** *Assuming conditional on the event that there is no change of the environment, under Assumptions 1 to 3, it holds that the interval between $\underline{t_k}$ is always increasing, say $\underline{t_{k+1}} - \underline{t_k} \geq \underline{t_k} - \underline{t_{k-1}}$. Meanwhile, it holds that the optimal policy re-solves at most $\min\{T, \frac{\log(\rho C/(\rho C+L-U))}{\log(U/L)}\} \lesssim \mathcal{O}(\frac{1}{C})$ times.*

We also observe the same phenomenon in our experiments. In Figure 1, we see that the learned policy gradually increases the re-solving interval between two change points. This is due to the diminishing marginal effect of additional observations. Theorem 3 further corroborates the superiority of our framework. Florence et al. (2025) also gives an upper bound $T - \Theta(\sqrt{C})$ of the re-solving time. However, our bound is much tighter. As we only need to re-solve constant times when the time horizon $T$ goes to infinity, they can only avoid re-solving in constant time horizons.

Together with Corollary 2, we know that the upper bounds of the optimal re-solving times show double phase transitions (cf. Figure 6 in Appendix B.5). Define $f(x) = \frac{\log(\rho x/(\rho x+L-U))}{\log(U/L)}$. When $C \leq f^{-1}(T)$, the upper bound is $T$. The upper bound becomes $\frac{\log(\rho C/(\rho C+L-U))}{\log(U/L)}$ when $f^{-1}(T) \leq C \leq \frac{U}{\rho}$. Finally, when $C \geq \frac{U}{\rho}$, the upper bound of re-solves ends up as 1.

We immediately obtain the following corollary when there are at most $N - 1$ change points which divide $T$ horizons into $N$ periods.

**Corollary 3.** *When there are at most $N$ distinct periods over the entire horizon, the optimal policy re-solves at most $\min\{T, \frac{N \log(\rho C/(\rho C+L-U))}{\log(U/L)}\} \lesssim \mathcal{O}(\frac{N}{C})$ times.*

In general, changes in the environment are accompanied by re-solving. Therefore, as $N$ increases, the upper bound on the number of re-solvings also grows.

## 4 METHODOLOGY

This section outlines the methods for constructing the POC framework, as shown in Figure 2.

**Data selection.**    Since the optimal online solution is to set the objective to $\mathbb{E}_{p_t}[c_t]$, we use the sample mean derived in the past to estimate it. However, one difficulty lies in the fact that $c_1, ..., c_{t-1}$ may be generated from different distributions. Thus, it is necessary to identify the informative samples, namely those originating from the same distribution. Our approach begins by applying a change point detection (CPD) algorithm to locate the most recent change point prior to time $t$. The data following this change point are then used to estimate $p_t$. We compare change point detection with other methods, like exponential moving averages (Hunter, 1986) and sliding windows (Datar et al., 2002), in Appendix D.6. Maillard (2019) proves that the detection delay is at most $\mathcal{O}(\frac{\log T}{\Delta^2})$ where $\Delta$ is the change magnitude. Notice that the impact of delay and that of estimation are of the same order (Wainwright, 2019), so change point detection will not be the main source of our cumulative loss. In our experiments, we employ a random-forest–based change point detector (Londschien et al., 2023), as it is relatively robust. Designing specialized change point detectors for different data scenarios is a worthwhile direction for future research.

**Feature engineering.**    Motivated by Assumption 2, we seek to construct features that capture the information set $\mathcal{I}_t$. Recall that re-solving becomes necessary in two situations, say when the environment experiences a significant shift, or when an increase in observations enhances the accuracy of our estimate of the environment. For the first case, inspired by linear programming, we assume that $\lambda$ and $\mu$ are the slack variables corresponding to the constraints $Ax \leq b$ and $x \geq 0$ of the relaxation of the MILP, respectively. We choose the gradient $c - A^T\lambda + \mu$ of the corresponding Lagrangian function as a feature, which reflects the suboptimality of the old solution under the new MILP. In addition, we incorporate $\lambda$ and $\mu$, which reveal which constraints are binding. Note that the first step in solving an MILP usually involves solving its linear relaxation, so the construction of features does not incur additional computational cost. In the second case, we use as features both the number of samples used by the previous solution to estimate the environment and the number of samples that would be used if re-solving were performed at the current time. These features quantify the extent to which additional samples improve the quality of the solution. We also record the solution's age to capture the probability of environmental change, due to Assumption 1. Moreover, following Theorem 1, we approximate the infinite-horizon re-solving problem using relative time. We detail other auxiliary features and conduct an ablation study on the features' effectiveness in Appendix D.6.

**Policy learning.**    Through feature engineering, denoted as $s_t(\mathcal{I}_t)$, we use an actor network to generate actions. We first sample from offline data to learn the policy. Since in practice $s_t$ does not form a perfect MDP, we choose on-policy RL, which concentrates on policy learning, rather than model-based RL, which learns the reward and transition, to enhance robustness (Janner et al., 2019). We use $\theta$ to denote the parameters in both the action policy $\pi_\theta$ and the value function $V_\theta$. Recall Theorem 1, in practice, the frequency of unknown environment changes is much lower compared to the data flow, so we assume $\rho \approx 0$ in the reward. This slightly reduces the weight of the re-solving cost and strengthens exploration in offline learning. For the discount factor $1 - \rho$, we approximate it with a single hyperparameter $\gamma$. Here, we adopt a moderate value of $\gamma$ to reduce the variance of the value function caused by error accumulation (Munos, 2003). Accordingly, during the sampling phase, we log the value estimate $V_\theta(s_t)$, the action $\xi_t \sim \pi_\theta^\xi(s_t)$, and the negative loss $c_t x_t^* - c_t x_t - C\xi_t$. Subsequently, we compute the TD-based advantage $A_t = c_t x_t^* - c_t x_t - C\xi_t + \gamma V_\theta(s_{t+1}) - V_\theta(s_t)$, and record the action probability $\pi_\theta^\xi(\xi_t|s_t)$ at time $t$. Therefore, after collecting the offline samples, we tailor proximal policy optimization (PPO) (Schulman et al., 2017) and update the neural network to obtain a new policy,

$$\theta' \leftarrow \arg\min_{\theta'} L(\theta')$$

$$:= \mathbb{E}_t \left[ -\min\{\frac{\pi_{\theta'}^\xi(\xi_t|s_t)}{\pi_\theta^\xi(\xi_t|s_t)} A_t, \text{Clip}(\frac{\pi_{\theta'}^\xi(\xi_t|s_t)}{\pi_\theta^\xi(\xi_t|s_t)}, 1 - \epsilon, 1 + \epsilon)A_t\} + c_1(V_{\theta'}(s_t) - A_t - V_\theta(s_t))^2 - c_2 H[\pi_{\theta'}^\xi](s_t) \right],$$

where $H$ is the policy entropy of the new policy, and $\epsilon$, $c_1$ and $c_2$ are hyperparameters. We outline our offline and online POC framework in Algorithms 1 and 2 in Appendix C.1.

In practice, once a sufficient number of online samples have been collected, they can be used to fine-tune the actor network, thereby further improving the model's performance.

## 5 EXPERIMENTS

Prior literature has largely lacked dynamic MILP datasets, in which the loss with respect to the objective exhibits non-smooth variations. Instead, most existing studies on cost-aware retraining have primarily focused on relatively simple regression or classification tasks (Mahadevan & Mathioudakis, 2023; Hoffman et al., 2024; Florence et al., 2025). To systematically investigate the question of when to re-solve, we construct a benchmark encompassing diverse classes of MILPs. Our experiments examine eight MILP families, highlighting the performance advantages of the POC framework relative to other algorithms when re-solving costs are present. In addition, we provide an analysis of how varying levels of re-solving cost influence both the frequency of re-solving decisions and the resulting cumulative loss.

### 5.1 DATASETS AND BASELINES

We present results on synthetic and real datasets. We design five synthetic datasets, including Set Cover (SC), Matching (Mat), Set Packing (SP), Facility Location (FL), and general MILP (GMILP). In addition, we curate a real-time second-level GTFS dataset for the New York area (Antrim et al., 2013; McHugh, 2013) from the Microsoft Fabric platform to construct two MILP families, namely Shortest Path Problem (SPP)[1] and Travelling Salesman Problem (TSP). Furthermore, we obtain Combinatorial Auction (CA) data from Leyton-Brown et al. (2000) and built a dynamic auction dataset. This provides a foundation, say a comprehensive set of both synthetic and real-world data, for future research on optimization problem scheduling. In Appendix D.1, we detail the specific optimization formulations of these various MILP families.

In what follows, we describe the baseline methods considered in our study. For the change point detector `CPD`, which returns the latest change point, we adopt a unified approach by employing random forests (Londschien et al., 2023) across all methods to ensure a fair comparison of different algorithms for deciding when to re-solve. To the best of our knowledge, this is the first work that systematically studies when to re-solve MILPs. Therefore, we tailor existing algorithms originally developed for determining when to retrain predictive models and fine-tune their hyperparameters to serve as baselines.

Specifically, **ADWIN-5%** (Bifet & Gavalda, 2007) triggers re-solving when a fundamentally different new change point is identified with 95% confidence, while **CARA-P** (Mahadevan & Mathioudakis, 2023) periodically solves the MILPs. **UPF**, proposed by Florence et al. (2025), decides the re-solving timing by predicting the uncertainty of future performance. In addition, Mahadevan & Mathioudakis (2023) introduces another two algorithms, CARA-T and CARA-CT, which rely on knowledge of future objectives and their similarity to historical data, which is intractable in our setting. Nevertheless, inspired by their definition of similarity, we augment UPF with a similarity-based feature, which improves upon the performance of the original UPF. We detail their implementations in Appendix D.4 and will analyze in the next section why these algorithms exhibit certain limitations in the context of deciding when to re-solve MILPs.

### 5.2 RESULTS

In the first experiment, we fix the re-solving cost at $C = 10$. Across eight synthetic and real-world datasets, our POC framework consistently outperforms the baseline algorithms, reducing the cumulative loss by an average of 2% to 17%. We list the cumulative loss (CL) and the total re-solving times (# R-S) of different algorithms in Table 1. Moreover, in high-frequency data scenarios, our POC framework substantially decreases the number of re-solving events, reducing the re-solving frequency to below 5%. In addition, we provide two lower bounds for the cumulative loss. First, we derive a lower bound under the assumption of knowing the change points in advance (LBwCP), i.e., if we had an oracle that revealed the exact locations of the change points and we re-solved every time ignoring the re-solving cost, what the cumulative optimization loss would be. This characterizes the unavoidable loss induced by the inherent randomness of the environment. Second, we derive a lower bound without knowing the change points (LBwoCP). At each time step, it employs `CPD` for data selection and updates the MILP solution without accounting for the re-solving cost. This measures

---

[1]SPP is a P problem, while the other seven families are NP-hard. For the sake of comparability, we also formulate SPP as a MILP; however, modern solvers can typically solve it very efficiently.

Table 1: Experimental Results: We report the cumulative loss and the number of re-solving events for all algorithms on eight datasets. In the tables, the best results are highlighted in **bold**, and the second-best results are underlined.

| **Algo** | SC | | Mat | | SP | | FL | |
|---|---|---|---|---|---|---|---|---|
| | CL ($\downarrow$) | # R-S | CL ($\downarrow$) | # R-S | CL ($\downarrow$) | # R-S | CL ($\downarrow$) | # R-S |
| ADWIN-5% | 2046.06 | 63.60 | 2473.62 | 53.20 | 2872.67 | 51.10 | 3009.56 | 55.10 |
| CARA-P | 2845.04 | 67.00 | 2850.80 | 91.00 | 3591.99 | 67.00 | 3501.49 | 63.00 |
| UPF | 8562.69 | 720.10 | 9086.74 | 724.00 | 9396.33 | 721.90 | 9889.30 | 755.80 |
| POC (ours) | **1771.78** | 22.00 | **2423.56** | 24.10 | **2815.81** | 26.50 | **2646.49** | 14.60 |
| LBwCP | 971.24 | - | 1349.15 | - | 1826.46 | - | 2081.33 | - |
| LBwoCP | 1360.52 | - | 1845.83 | - | 2176.54 | - | 2332.99 | - |
| **Algo** | GMILP | | SPP | | TSP | | CA | |
| | CL ($\downarrow$) | # R-S | CL ($\downarrow$) | # R-S | CL ($\downarrow$) | # R-S | CL ($\downarrow$) | # R-S |
| ADWIN-5% | 2747.42 | 74.10 | 968.94 | 15.00 | 1565.95 | 11.00 | 8375.44 | 178.30 |
| CARA-P | 3243.08 | 67.00 | 857.77 | 15.00 | 1244.86 | 15.00 | 6337.09 | 42.00 |
| UPF | 7030.22 | 517.00 | 1336.47 | 61.00 | 4337.65 | 2.00 | 10352.47 | 308.70 |
| POC (ours) | **2280.09** | 18.90 | **825.96** | 15.50 | **1161.10** | 13.70 | **5736.52** | 16.50 |
| LBwCP | 1631.09 | - | - | - | - | - | - | - |
| LBwoCP | 1859.58 | - | 546.23 | - | 837.37 | - | 5464.87 | - |

the additional unavoidable loss introduced by the choice of change point detector. Therefore, the gap between the cumulative loss and LBwoCP represents the potential loss incurred by a strategy due to the presence of re-solving costs. For our POC framework, we provide the following remark.

**Remark 1.** *For our POC framework, the re-solving cost within the cumulative loss is roughly comparable to the actual optimization loss minus the unavoidable optimization loss, i.e., LBwoCP.*

Remark 1 suggests that, in the presence of re-solving costs, it is necessary to reduce the frequency of re-solving. The optimal policy must balance the re-solving cost against the increase in optimization loss that arises from less frequent re-solving. This provides a fundamental guideline for addressing this class of decision-making problems.

In the second experiment, we vary the re-solving cost $C$ from 5 to 50 and examine the performance of different algorithms under distinct costs on the general MILP dataset, as shown in Figures 3 and 4. Unsurprisingly, we find that as $C$ increases, the number of re-solving events decreases while the cumulative loss rises. Notably, ADWIN-5% tends to re-solve shortly after detecting a new change point. Although it has been fine-tuned, it nevertheless remains largely insensitive to variations in the re-solving cost. We also find that our POC framework exhibits strong robustness to misspecified re-solving costs in Figure 5. Even when provided with an incorrect cost parameter, POC continues to perform well. For instance, when the provided $C$ is as large as 50, ten times the true cost of 5, the performance of POC decreases by less than 25%, and on average, the degradation is below 5%. In practice, the tradeoff between re-solving cost and optimization loss is often difficult to estimate accurately. Thus, the robustness demonstrated by the POC framework highlights its broad potential for practical deployment.

Finally, we discuss why the baseline algorithms perform poorly on the task of determining when to re-solve. We first note the non-negligible gap between LBwCP and LBwoCP. Change point detectors typically identify a change only some time after it has actually occurred. Moreover, CPD is primarily designed for offline data, and in our online setting, it's prone to detecting spurious change points when the sample size is small. In contrast to ADWIN-5%, when POC encounters such false change points, it refrains from triggering a re-solve due to the limited data available and the resulting inaccuracy in environment estimation. Consequently, POC achieves fewer re-solving events and lower cumulative loss. For CARA-P, since the distribution of change points is highly irregular, we are forced to shorten the period to avoid missing essential change points. This, however, introduces a large number of unnecessary re-solving events. UPF, on the other hand, tends to suffer from error explosion under high-speed data streams. The original UPF is designed for $T = 8$, whereas our six datasets have $T = 1000$ and the other two have $T = 300$ during the test phase. In such settings, even small errors can accumulate to exceed the magnitude of $C$, rendering model-based approaches ineffective. Moreover, the original UPF requires enumerating all possible

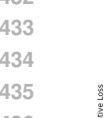 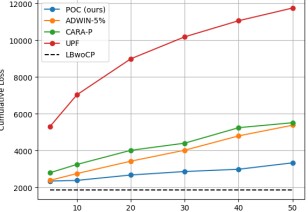 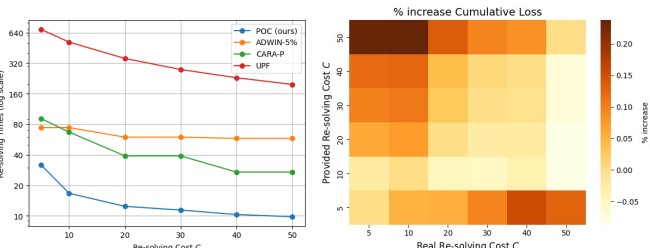

Figure 3: POC consistently reaches low cumulative loss across distinct re-solving costs.

Figure 4: Log-scale total re-solves of different algorithms across distinct re-solving costs.

Figure 5: Cumulative loss sensitivity to misspecified re-solving cost in percentage increase.

re-solving time points, which grows exponentially with $T$ and is therefore impractical for large-scale data. These observations demonstrate the substantial advantage of our direct policy-learning framework in high-frequency environments.

### 5.3 Design Choices Analysis and Ablation Study

Compared to MILP, linear programming can be solved much more efficiently. We therefore experiment with using linear programming to pre-train the policy, followed by MILP to fine-tune the learned actor network. We find that this approach substantially reduces the number of epochs required for policy learning, with only about a 2% performance loss. In addition, we evaluate the framework under different discount factors, network architectures, and data selection strategies, thereby providing a practical recipe for applying the POC framework in real-world settings. The corresponding results are reported in Appendix D.6.

Compared to previous work, we additionally consider the beneficial sample size as a feature. Re-solving can be triggered either by a detected essential change point or by the accumulation of sufficient observations that enable more accurate estimation of the environment. Theorem 3 provides theoretical guarantees, and our ablation study shows that incorporating beneficial sample size reduces cumulative loss by roughly 15% with a slight increase in re-solving frequency (cf. Table 11 in Appendix D.6). These results highlight the value of beneficial sample size in guiding more effective re-solving decisions.

## 6 Conclusion and Discussion

We propose a practical formulation of the fundamental problem of determining when to re-solve dynamic MILPs. Unlike internal adaptivity, which adjusts re-solving time solely based on model performance, our setting also accounts for external costs, where the re-solving time is determined by external adaptivity. Our theoretical analysis characterizes the structural properties of re-solves and motivates the design of the POC framework for policy learning. Building on these insights, we introduce a robust POC framework, which integrates data selection, feature engineering, and policy optimization. Across eight benchmark datasets, POC consistently outperforms strong baselines, reducing cumulative loss by 2% to 17%. These results highlight the effectiveness of incorporating re-solve-aware features into policy design and establish a policy-based rather than model-based paradigm for real-time operations in high-frequency data streams.

Having put forward this overlooked problem, several questions naturally arise for future exploration. For nonlinear optimization, how should one characterize the suboptimality of outdated solutions? When the feature space is large or networks are deep, for instance, in optimization over text or image domains, how do other policy learning algorithms, such as group relative policy optimization (Shao et al., 2024), compare in terms of efficiency and robustness? Relative to full re-solving, how might one combine our approach with efficient optimization techniques such as subproblem selection (Li et al., 2025)? Finally, how can the POC framework be extended to broader cost-aware decision-making settings, such as tool use in large language models (Wang et al., 2025)? We leave these directions as promising avenues for future research.

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

# A   OMITTED DETAILS IN SECTION 1

## A.1   RELATED WORKS

We summarize below three lines of existing literature pertinent to our work.

**Change point detection.**   Change point detection is a widely studied topic in statistics. We use a change point detector to filter objectives from the same distribution and leverage them to estimate the environment and construct part of the strategy for generating new re-solving solutions. Aminikhanghahi & Cook (2017) provides a survey of traditional statistical methods for detecting change points. In recent years, a number of papers have explored the application of deep learning to change point detection. Londschien et al. (2023) establishes theoretical guarantees for detecting change points using random forests, while Li et al. (2024a) demonstrates the relationship between multilayer perceptrons (MLPs) and change point detection statistics. Dmitrienko et al. (2022) discusses the advantages of transformer architectures in change point detection, and with the rapid development of large language models (LLMs), Dong et al. (2024) investigates whether LLMs can be used for anomaly detection. Interested readers may refer to Xu et al. (2025), which provides a review of various deep learning algorithms for change point detection and discusses the prospects of integrating CPD with decision-making. Meanwhile, Van den Burg & Williams (2020) constructs a manually annotated dataset and a metric for comparing existing CPD algorithms. Maillard (2019) investigates the relationship between the detection delay of a change point and the magnitude of the change under sub-Gaussian distributions. Wei & Srivatsva (2018); Besson et al. (2022) study bandit problems with change points and establish regret bounds. Bifet & Gavalda (2007); Pesaranghader & Viktor (2016); Raab et al. (2020) directly treat the detection of a distribution shift as a signal to change the policy. None of them accounts for the cost of policy changes; nevertheless, we aim not only to detect change points but also to identify meaningful changes for re-optimization. Our algorithm hence demonstrates significant advantages in scenarios with high re-solving costs.

**Re-solving scheduling.**   Few papers have addressed the practical question of when to re-solve a costly MILP problem under an uncertain environment. Meignan (2014); Schieber et al. (2018) use the bound on the distance between the old and new solutions to model the cost of re-solving. Yuan et al. (2022), on the other hand, does not account for the re-solving cost and instead focuses solely on improving the solution method. Zych (2012) uses the solution from one instance to approximate another NP-hard instance, and so does Mannino & Sartor (2020). However, our paper focuses on deciding when to re-solve and treats the solver as a black box. Our setup is also related to recent work on determining when to retrain a machine learning model. Mahadevan & Mathioudakis (2023) uses a heuristic algorithm to decide when to retrain a classifier, while Hoffman et al. (2024) uses a heuristic algorithm to address the corresponding regression problem. Unlike in regression, applying their calibration method to an MILP can easily violate constraints and yield an infeasible new solution, which introduces additional challenges to the problem we study. Florence et al. (2025) quantifies uncertainty to decide when to retrain a binary classifier. They consider a low-data setting, with a time horizon $T$ of only 8. As a result, they can enumerate the action space to obtain the ground truth. In contrast, we consider high-frequency data streams from real-world data platforms with a very large $T$, where their model-based supervised approach is not applicable. Instead, we adopt a model-free on-policy method to address this problem.

**Reinforcement learning for varying environments.**   There is a vast amount of literature on non-stationary reinforcement learning (Padakandla et al., 2020; Mao et al., 2020; Padakandla, 2021; Dmitrienko et al., 2022; Cheung et al., 2023). They do not account for the cost of changing the policy, whereas the cost associated with re-solving makes our problem non-smooth. Baumann et al. (2018) uses deep RL to study event-triggered control with fixed dynamics. But since we need to select data to estimate the future, our scenario is not a Markovian decision process, which makes vanilla MDP-based RL algorithms not directly applicable. This issue also arises when using RL for various types of scheduling (Zhang & Dietterich, 1995; Mao et al., 2016; Waschneck et al., 2018; Zhang et al., 2020a; Liu et al., 2020; Shyalika et al., 2020; Liu et al., 2022). To address this issue, we do not directly use the changing optimization goal as the state. Instead, we employ feature engineering to mitigate the effects of the non-MDP nature of the problem. Moreover, rather than estimating model-based quantities such as transitions, we adopt a less sensitive policy-learning approach to

directly determine the re-solving times. We tailor proximal policy optimization (Schulman et al., 2017) to learn the policy; however, most policy-learning algorithms of this kind (Williams, 1992; Sutton et al., 1999; Schulman et al., 2015; Mnih et al., 2016) rely on a discount factor, whereas we consider infinite-horizon learning without discounting. We establish an equivalence between the changing environment and the discount factor, thereby theoretically justifying the success of our POC framework.

# B    OMITTED PROOF IN SECTION 3

## B.1    PROOF OF THEOREM 1

With a little abuse of notations, we use $\pi^*$ to represent the best action facing $s_t$. We assume that if the environment doesn't change over time, say $p_t = p_{t-1}$, it holds that $s_{t+1} \sim \mathbb{P}^*(\cdot \mid s_t)$. From Assumption 1, we assume that the expected optimization loss of old solution in a new environment is $R$. Therefore, it holds that

$$V^*(s_t) = \mathbb{E}_{c_t \sim \mathbb{P}^*(c_t \mid s_t)}[(1-\rho)(c_t^T x_t^* - c_t^* x_t) - C\xi_t] + \rho R + (1-\rho)\mathbb{E}_{s_{t+1} \sim \mathbb{P}^*(s_{t+1} \mid s_t)}V^*(s_{t+1}) + \rho \mathbb{E}V^*(s_{t+1}).$$

Recall that for every policy $\pi$, we have $\xi_t = \pi^\xi(s_t)$ and $x_t = \pi^x(s_t)$. For the last term, since when the environment changes, the change can be arbitrary, it's the same as a restart from time 1 for our infinite-horizon decision-making problem. From Assumption 1, we know that this term is independent of state $s_t$. Hence, we only need to find optimal strategy

$$\pi^* = \arg\max_{\pi=(\pi^\xi, \pi^x)} \mathbb{E}_{c_t \sim \mathbb{P}^*(c_t \mid s_t)}[(1-\rho)(c_t^T x_t^* - c_t^* x_t) - C\xi_t] + (1-\rho)\mathbb{E}_{s_{t+1} \sim \mathbb{P}^*(s_{t+1} \mid s_t)}V^*(s_{t+1}),$$

as $R$ is unrelated with $\pi^*$. Besides, since when the environment changes, $s_{t+1}$ is a new start with the same dynamic, for example, the probability of environment changes is still $\rho$, maximizing $\mathbb{E}_{c_t \sim \mathbb{P}^*(c_t \mid s_t)}[(1-\rho)(c_t^T x_t^* - c_t^* x_t) - C\xi_t] + (1-\rho)\mathbb{E}_{s_{t+1} \sim \mathbb{P}^*(s_{t+1} \mid s_t)}V^*(s_{t+1})$ will optimize $\rho\mathbb{E}V^*(s_{t+1})$ naturally.

We apply the above procedure recursively. It holds that the optimal strategy satisfies

$$\pi^* = \arg\max_{\pi=(\pi^\xi, \pi^x)} \sum_{\tau=t}^{T} \mathbb{E}_{\mathbb{P}^*(\cdot \mid s_\tau)}[(1-\rho)^{\tau-t}((1-\rho)(c_\tau^T x_\tau^* - c_\tau^T x_\tau) - C\xi_\tau)].$$

Therefore, we know that finding $\pi^*$ with respect to $V^*$ is equivalent to find $\pi^*$ with respect to a new value function $\widetilde{V}^*$ with discount rate $1-\rho$ and reward $(1-\rho)(c_t^T x_t^* - c_t^* x_t) - C\xi_t$, which shows the equivalence of $\widetilde{V}^*$ and $V^*$. Note that the transitional kernel $\mathbb{P}^*(\cdot|s_t)$ is now slightly different from the true environment, as we assume the environment doesn't change over time. This design is only for analysis. In practice, this formulation is also beneficial to policy learning. As the environment change brings high variance to the learning process, focusing on the unchanged part will stabilize our learning process, which motivates our design in the region of high data and low switching.

## B.2    PROOF OF THEOREM 2

Before proving Theorem 2, we first give an example under which Assumption 3 holds.

**Example 1.** *Within a stable environment, when the solution $x$ is bounded, it holds that $\mathbb{E}[c_t^T x_t - c_t^T x_t^*] \lesssim \mathcal{O}(n^{-\frac{1}{2}})$ using $n$ observations to estimate $c_t$.*

*Proof.* Since we use $n$ i.i.d. observations to estimate $\mathbb{E}[c_t]$, we know that the mean square error $\Delta$ is less than $\mathcal{O}(\frac{1}{\sqrt{n}})$ (Wainwright, 2019). Since $x$ is bounded, we assume that $\|x\|_2 \leq B$. Let's now consider two optimization problems, say $\min_x c_i^T x$ subject to $Ax \leq b$ and $x \in \mathbb{Z}_{\geq 0}$. It then holds that

$$\begin{aligned} c_2^T x_1^* - c_2^T x_2^* &= c_2^T x_1^* - c_1^T x_1^* + c_1^T x_1^* - c_2^T x_2^* \\ &\leq c_2^T x_1^* - c_1^T x_1^* + c_1^T x_2^* - c_2^T x_2^* \\ &\leq \|c_2 - c_1\|_2 \cdot (\|x_1^*\|_2 + \|x_2^*\|_2) \\ &\leq 2B\|c_2 - c_1\|_2. \end{aligned}$$

Here, we use the fact that $x_1^*$ is the optimal solution to the first MILP, so $c_1^T x_1^* \leq c_1^T x_2^*$. Then, as we have $\Delta \lesssim \mathcal{O}(\frac{1}{\sqrt{n}})$, it immediately yields

$$\mathbb{E}[c_t^T x_t - c_t^T x_t^*] \leq 2B\Delta \lesssim \mathcal{O}(n^{-\frac{1}{2}}),$$

which ends the proof. So, we know that the order of the upper bound is at least $\frac{1}{2}$. $\qquad\square$

We now turn to Theorem 2. With the help of Theorem 1, we know that conditional on a stable environment, we only need to consider a $(1-\rho)$-discount MDP. We use $\gamma$ to denote $1-\rho$. We re-solve the MILP at time $t_k$, and we assume that we wait $n = t_{k+1} - t_k$ to re-solve another time. Therefore, we have an extra re-solving cost $\gamma^n C$ after discounting. The optimization loss only differs after $t_{k+1}$, so the gap is at most $(\gamma^n + \gamma^{n+1} + \cdots)(U(t_k - 1)^{-\alpha} - L(t_k + n - 1)^{-\alpha})$ due to Assumption 3. Recall that when we use the $\gamma$-discount MDP, the reward becomes $\gamma(c_t^T x_t^* - c_t^T x_t) - C\xi_t$, and we should adjust the optimization loss considering discount as well.

Therefore, to motivate the re-solve at $t_{k+1}$, we have

$$\gamma^n C \leq \frac{\gamma^n}{1-\gamma}(U(t_k - 1)^{-\alpha} - L(t_k + n - 1)^{-\alpha}).$$

It yields that for the optimal strategy, the interval should satisfy

$$n \geq 1 - t_k^* + [\frac{L}{U(t_k^* - 1)^{-\alpha} - (1-\gamma)C}]^{1/\alpha}.$$

Thus, we know that

$$t_{k+1}^* \geq 1 + [\frac{L}{U(t_k^* - 1)^{-\alpha} - (1-\gamma)C}]^{1/\alpha},$$

which finishes our proof.

### B.3 PROOF OF COROLLARY 1

We know that $t_{k+1}^* \geq 1 + [\frac{L}{U(t_k^*-1)^{-\alpha}-(1-\gamma)C}]^{1/\alpha}$ and $1 + [\frac{L}{U(t_k^*-1)^{-\alpha}-(1-\gamma)C}]^{1/\alpha}$ is increasing with $t_k^*$. Assuming $t_k^*$ has a lower bound $\underline{t_k}$, i.e., $t_k^* \geq \underline{t_k}$, it holds that

$$t_{k+1}^* \geq 1 + [\frac{L}{U(t_k^* - 1)^{-\alpha} - (1-\gamma)C}]^{1/\alpha} \geq 1 + [\frac{L}{U(\underline{t_k} - 1)^{-\alpha} - (1-\gamma)C}]^{1/\alpha} = \underline{t_{k+1}},$$

which ends the proof.

### B.4 PROOF OF COROLLARY 2

Note that $t_1^* = 2$, therefore, for the proof of Theorem 2, when $\gamma^n C \geq \frac{\gamma^n}{1-\gamma}(U * 1^{-\alpha} - L(n+1)^{-\alpha})$ for all $n$, we would never re-solve the MILP. It's equivalent to $C \geq \frac{U}{\rho}$, which ends the proof.

### B.5 PROOF OF THEOREM 3

From Corollary 1, we know that $\underline{t_{k+1}} = 1 + [\frac{L}{U(\underline{t_k}-1)^{-\alpha}-(1-\gamma)C}]^{1/\alpha}$. Let's construct a numerical sequence $x_k = (\underline{t_k} - 1)^{-\alpha}$, and $x_1 = 1$. Since $\underline{t_k}$ increases with $k$, we have that $x_k$ should be decreasing. In the meantime, we have that $x_{k+1} = \frac{U}{L}x_k - \frac{(1-\gamma)C}{L}$. Therefore, it holds that

$$x_{k+1} - \frac{(1-\gamma)C}{U-L} = \frac{U}{L}(x_k - \frac{(1-\gamma)C}{U-L}).$$

Since $x_1 = 1$, we know that

$$x_k = (\frac{U}{L})^{k-1}(\frac{U-L-(1-\gamma)C}{U-L}) + \frac{(1-\gamma)C}{U-L}.$$

Note that $x_2 = \frac{U-(1-\gamma)C}{L}$. We need $x_2 \leq x_1$, which is equivalent to $C \geq \frac{U-L}{1-\gamma}$. Otherwise, we use a trivial upper bound $T$ to bound the number of re-solves. Since now, we only focus on the region that $C \geq \frac{U-L}{1-\gamma}$.

When $Ux_k \leq (1-\gamma)C$, we won't re-solve anymore due to the proof of Corollary 2. We then compute the largest $k$ such that $x_k \geq \frac{(1-\gamma)C}{U}$. It holds that

$$\arg\max_{k}\{k : (\frac{U}{L})^{k-1}(\frac{U-L-(1-\gamma)C}{U-L}) + \frac{(1-\gamma)C}{U-L} \geq \frac{(1-\gamma)C}{U}\}$$

$$= \frac{\log((1-\gamma)C)/((1-\gamma)C + L - U)}{\log(U/L)}.$$

So, we know that the optimal strategy re-solves at most $\frac{\log(\rho C)/(\rho C+L-U)}{\log(U/L)}$ times. We notice that the upper bound is independent of the decay rate $\alpha$. It arises from the fact that Assumption 3 supposes homogeneous decay over time. When the decay rate falls into an interval, we can similarly obtain the recurrence formula for $x_k$, which now depends on the rate ratio, and solve the sequence. In practice, we can simulate and obtain the sequence; however, from a theoretical perspective, we may not be able to derive a closed form. Therefore, we retain Assumption 3 as it provides all essential insights into the re-solving times.

Finally, let's prove $t_{k+1} - t_k$ is increasing. Since $x_k = (\frac{U}{L})^{k-1}(\frac{U-L-(1-\gamma)C}{U-L}) + \frac{(1-\gamma)C}{U-L}$, we know that

$$t_k = 1 + [(\frac{U}{L})^{k-1}(\frac{U-L-(1-\gamma)C}{U-L}) + \frac{(1-\gamma)C}{U-L}]^{-1/\alpha}.$$

Denote $h(x) = 1 + [(\frac{U}{L})^{x-1}(\frac{U-L-(1-\gamma)C}{U-L}) + \frac{(1-\gamma)C}{U-L}]^{-1/\alpha}$. It's sufficient to prove that $h(x)$ is a convex function. It holds that

$$h'(x) = -\frac{1}{\alpha}[(\frac{U}{L})^{x-1}(\frac{U-L-(1-\gamma)C}{U-L}) + \frac{(1-\gamma)C}{U-L}]^{-1/\alpha-1}(\frac{U-L-(1-\gamma)C}{U-L})(\frac{U}{L})^{x-1}\log(\frac{U}{L}).$$

Since $\alpha$ is positive, we know that $-\frac{1}{\alpha} \leq 0$. Meanwhile, as $U \geq L$, we know that $\log(\frac{U}{L}) \geq 0$ and $(\frac{U}{L})^{x-1}$ is increasing. Additionally, since $C \geq \frac{U-L}{1-\gamma}$, it holds that $\frac{U-L-(1-\gamma)C}{U-L} \leq 0$, and $[(\frac{U}{L})^{x-1}(\frac{U-L-(1-\gamma)C}{U-L}) + \frac{(1-\gamma)C}{U-L}]^{-1/\alpha-1}$ is increasing. Therefore, we know that $h'(x)$ is an increasing function, which means that $h(x)$ is convex. We then obtain that $t_{k+1} - t_k$ is increasing, or in other words, $t_{k+1} - t_k \geq t_k - t_{k-1}$.

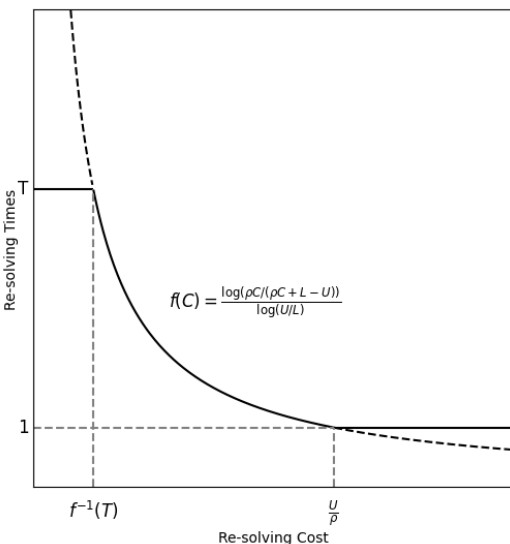

Figure 6: Upper bound of the optimal re-solving times: We observe double phase transitions in the upper bound of the optimal re-solves. The optimal number of re-solving times should always lie below the black solid line.

Together with Corollary 2, we can bound the number of re-solves for different re-solving costs and observe a double phase transition as shown in Figure 6. It holds that the optimal re-solving times

$\|\xi^*\|_1$ satisfies

$$\|\xi^*\|_1 \leq \begin{cases} T, & C \leq f^{-1}(T); \\ \frac{\log(\rho C/(\rho C+L-U))}{\log(U/L)}, & C \in [f^{-1}(T), \frac{U}{\rho}]; \\ 1, & C \geq \frac{U}{\rho}, \end{cases}$$

where $f(x) = \frac{\log(\rho x/(\rho x+L-U))}{\log(U/L)}$. Moreover, since it holds that $\frac{\log(\rho C/(\rho C+L-U))}{\log(U/L)} \leq \frac{U-L}{\log(U/L)(\rho C+L-U)}$, we know that $\|\xi^*\|_1 \lesssim \mathcal{O}(\frac{1}{C})$ as well. It means that the optimal number of re-solves decays at least as fast as $\mathcal{O}(\frac{1}{C})$ with respect to re-solving cost $C$. This finishes our proof.

### B.6 Proof of Corollary 3

We know for Theorem 3 that when there is no change point, the optimal strategy only needs to re-solve at most $\frac{\log(\rho C/(\rho C+L-U))}{\log(U/L)}$ times. Since Assumption 2 supposes that state $s_t$ can capture all useful information, including the location of the change points, we only need to restart at every change point. Therefore, we only need to re-solve at most $\frac{N \log(\rho C/(\rho C+L-U))}{\log(U/L)} \lesssim \mathcal{O}(\frac{N}{C})$ with $N-1$ change points. As the number of re-solves has a trivial upper bound $T$, it finishes our proof of Corollary 3 that the optimal policy re-solves at most $\min\{T, \frac{N \log(\rho C/(\rho C+L-U))}{\log(U/L)}\}$ times.

## C Omitted Details in Section 4

### C.1 Our Offline and Online POC Frameworks

---

**Algorithm 1** Offline POC Framework.

---
**Input:** Offline data $\{g_i\}_{i \in [I]}$, policy $\pi_\theta$, change point detector CPD.
**Initialization:** $g \leftarrow \emptyset$.
**for** $i \in [I]$ **do**
  **for** $t \in [T_i]$ **do**
    Observe $A_i$, $b_i$ and $c_{i1}, ..., c_{i(t-1)}$.
    Select data starting at $\iota = \text{CPD}(c_{i1}, ..., c_{i(t-1)})$.
    Construct feature $s_{it}$.
    Choose re-solving action $\xi_{it} \sim \pi_\theta^\xi(s_{it})$.
    **if** $\xi_{it} = 1$ **then**
      Solve $x_{it} = \arg\min_x \frac{\sum_{j=\iota}^{t-1} c_{ij}^T x}{t-\iota}$ subject to $A_i x \leq b_i$ and $x \in \mathbb{Z}_{\geq 0}$, and update solution.
    **end if**
    Observe $c_{it}$ and computer advantage $A_{it}$.
    Update $g \leftarrow g \cup (s_{it}, \xi_{it}, V_\theta(s_{it}), A_{it}, \pi_\theta^\xi(\xi_{it}|s_{it}))$.
  **end for**
**end for**
Update policy $\theta \leftarrow \arg\min_\theta L(\theta)$ over dataset $g$.
**Output:** Updated policy $\pi_\theta$.

---

## D Omitted Details in Section 5

In this section, we provide a detailed description of our experimental design and the results obtained.

---

**Algorithm 2** Online POC Framework.

    **Input:** Policy $\pi_\theta$, change point detector CPD, default solution $x$.
    **for** $t \in [T]$ **do**
        Observe $A$, $b$ and $c_1, ..., c_{t-1}$.
        Select data starting at $\iota = \text{CPD}(c_1, ..., c_{t-1})$.
        Construct feature $s_t$.
        Choose re-solving action $\xi_t \sim \pi_\theta^\xi(s_t)$.
        **if** $\xi_t = 1$ **then**
            Solve $x_t = \arg\min_x \frac{\sum_{j=\iota}^{t-1} c_j^T x}{t - \iota}$ subject to $Ax \leq b$ and $x \in \mathbb{Z}_{\geq 0}$.
            Update solution $x \leftarrow x_t$.
        **else**
            Use old solution $x$.
        **end if**
    **end for**

---

### D.1 DATASETS

We first detail in this section the construction of our eight datasets. For Set Cover, Facility Location and general MILP, the optimization problem is

$$\min_x c^T x$$
$$s.t. \ Ax \geq b$$
$$x \in \mathbb{Z}_{\geq 0}.$$

For SC, the matrix $A$ encodes the coverage structure, where rows correspond to elements and columns to sets, with $A_{ij} = 1$ indicating that set $j$ covers element $i$. It's generated by assigning each constraint to be covered by a random number of sets, ensuring at least one coverage per constraint, and then adding about 5% additional random entries. The vector $b$ specifies the coverage requirements, typically with all entries being 1 to enforce that each element must be covered at least once. The cost vector $c$ assigns a nonnegative cost to each set, and the objective is to identify a subset of sets of minimum total cost that satisfies all coverage constraints. Meanwhile, for FL, $A$ denotes the customer–facility incidence matrix, where $A_{ij} = 1$ if customer $i$ can be served by facility $j$. It's generated by connecting each customer to each facility independently with some probability, and ensuring that every customer is connected to at least one facility. The constraint vector $b = 1$ requires that each customer be covered by at least one open facility, while the cost vector $c$ specifies the opening cost of each facility. The objective is to minimize the total cost of selected facilities subject to the coverage constraints. Another setting is to restrict $x$ to be a binary variable. These two formulations are equivalent, since the optimal solution must take values of either 0 or 1. Additionally, for GMILP, we consider some general covering problems. We no longer constrain $b = 1$ and let $b$ be any positive random number. In addition, unlike SC, where $A$ is a sparse matrix and $A_{ij}$ can be either 0 or 1, the matrix $A$ for GMILP is a dense matrix and every entry comes from a uniform distribution. We then select MILPs with at least one feasible solution to form the dataset.

On the other hand, for the Matching, Set Packing and Combinatorial Auction dataset, the MILP formulation is

$$\max_x c^T x$$
$$s.t. \ Ax \leq b$$
$$x \in \mathbb{Z}_{\geq 0}.$$

In the Mat dataset, the matrix $A$ represents the vertex–edge incidence structure, where $A_{ve} = 1$ if edge $e$ is incident to vertex $v$. It's generated by randomly sampling some distinct undirected edges between vertices, with each column corresponding to an edge that is incident to exactly two vertices. The vector $b$ enforces the matching constraints, typically with $b = 1$ to ensure that each vertex is incident to at most one selected edge. The cost vector $c$ assigns a weight to each edge, and the objective is to maximize the total weight of the chosen edges subject to the matching constraints. Similarly, for SP, $A$ indicates element–set membership, with $A_{ij} = 1$ if element $i$ is in set $j$. Here, $A$

is generated by assigning each set to cover a random subset of elements, and each column contains at least one nonzero entry. The constraint vector $b = 1$ ensures that each element can belong to at most one chosen set, while the cost vector $c$ assigns weights to sets. The goal is to maximize total weight. Finally, for CA, the matrix $A$ represents the bundle-item structure, where $A_{ij}$ means the $j$-th bundle contains the $i$-th item. We set $b = 1$, assuming each item can be allocated at most once. The objective $c$ is the bid, and the auctioneer aims to maximize the total revenue. For all these datasets, we transform them into a unified form $\min_x c^T x$ subject to $Ax \leq b$ and $x \in \mathbb{Z}_{\geq 0}$ by changing the signs of $A$, $b$, and $c$.

For the Shortest Path Problem, we have MILPs with a formulation

$$\min_x c^T x$$
$$s.t.\ Ax = b$$
$$x \in \mathbb{Z}_{\geq 0}.$$

Here, $A$ is the node–edge incidence matrix, where each column encodes the orientation of an edge. The vector $b$ specifies flow balance, with $b = 1$ at the source, $b = -1$ at the sink, and zeros elsewhere. The cost vector $c$ records edge travel times. Minimizing the objective yields the shortest path between the source and the sink.

For the Travelling Salesman Problem, we study an NP-hard variant of the traditional TSP problem. We assume to start at a location and return to the same location. However, we constrain that the route has to pass by some assigned locations $V$. For example, consider Amazon trucks that depart from a central consolidation warehouse each morning, deliver parcels to a set of regional distribution depots, and finally return to the warehouse. The company may determine the visiting order of depots dynamically based on real-time traffic conditions. The optimization problem becomes

$$\min_x c^T x$$
$$s.t.\ A_x x = 0$$
$$A_f f = b_f$$
$$\sum_{\{e : e \to v\}} x_e \geq 1 \text{ if } v \in V$$
$$f \leq |V| x$$
$$f \geq 0$$
$$x \in \mathbb{Z}_{\geq 0}.$$

For TSP, the vector $x$ indicates whether each directed edge is selected, while the continuous vector $f$ represents the auxiliary flow used to eliminate subtours and guarantee connectivity. Two incidence matrices are introduced here. The matrix $A_x$ is the node–edge incidence matrix applied to the edge variables $x$. It enforces degree balance, i.e., the number of selected incoming edges equals the number of selected outgoing edges at every node, encoded as the constraint $A_x x = b_x$ with $b_x = 0$. The second matrix, $A_f$, is an identical incidence structure applied to the flow variables $f$. It imposes flow conservation according to $A_f f = b_f$, where $b_f$ specifies supply and demand, say the root node provides $|V|$ units of flow, each must-visit node consumes one unit, and all other nodes are neutral. Together, $(A_x, b_x)$ guarantees edge-balance for the route, while $(A_f, b_f)$ ensures that the auxiliary flow establishes connectivity from the root to all must-visit nodes.

We now proceed to describe the construction of the datasets. For each offline dataset, we designate the last 20 to 10 instances as the validation set, and the final 10 instances as the online test set.

For SC, Mat, SP, FL and GMILP, we set $T = 1000$ horizons and randomly choose three change points within every time series. Hence, in the first experiment, we leverage the true change points for the training process. The periods separated by change points are associated with objectives $c_t$ drawn from different underlying distributions, while we assume $A$ and $b$ remain the same, reflecting the fact that in practice, objectives change more frequently than constraints. In every dataset, we have 100 variables and 50 constraints.

For SPP and TSP, we use the New York City GTFS transit dataset (Antrim et al., 2013; McHugh, 2013) from June 17 to June 30, 2025. To reduce variance, traffic conditions are estimated by averaging observations within 10-minute intervals. The dataset covers 14,095 locations, which we cluster into 100 regions using a $K$-Medoids model with Manhattan distance (Kaufman & Rousseeuw,

2009). For experiments, we focus on the largest connected component of the induced region-level network, which consists of 94 regions and 313 edges. We choose $T = 300$ in both datasets, so we segment long time series into short sequences of length 300. We observe that the shortest paths fluctuate frequently over time. For certain origin–destination pairs, the optimal path changes more than 200 times during the observation period, say 2 weeks. For the TSP dataset, we assume that the route must visit five designated nodes in addition to the starting point. Therefore, we have 313 variables in these two datasets, and corresponding constraints are given in their formulations.

For the CA setting, we use the Combinatorial Auction Test Suite (CATS) (Leyton-Brown et al., 2000) to generate bids that mimic real-world bidding behaviors. We consider 5 items, which results in 31 possible bundle bids. We set $T = 1000$ in this dataset. At each horizon, there is a probability of 0.5% that a change point occurs in the bidding distribution. The distributions are drawn from the $L_2$ and $L_4$ categories in CATS, which are introduced by Sandholm (2002), and the $L_6$ category, which is proposed by Fujishima et al. (1999). Note that the distributions before and after a change point may be identical, which we use to simulate non-substantive change points.

### D.2 FEATURE ENGINEERING

After identifying the final change point $\iota$ using `changeforest` (Londschien et al., 2023), we construct a set of features to predict the optimal re-solving time. We use the subscript old to denote the variables corresponding to the previous solution.

We first employ a set of features to record the lifetime of the solution and the amount of data used. These features reflect whether the previous solution has become outdated and whether the current number of observations is sufficient to substantially improve the accuracy of environment estimation.

- $t$ minus the acquisition time of the old solution;
- The number of observations available for estimating the objective, i.e., $t - \iota$;
- The number of observations used by the old solution.

For each solving instance, we obtain a solution $x$ together with the slack variables $\lambda$ and $\mu$ of the corresponding linear program. Since the Lagrangian function can be written as $L(x) = c^T x - \lambda^T A x + \mu^T x$, we use its gradient $c - A^T \lambda + \mu$ to assess how well the old solution fits the newly observed environment.

- The gradient of the old solution in the new environment $\frac{\sum_{j=\iota}^{t-1} c_j}{t-\iota} - A^T \lambda_{\text{old}} + \mu_{\text{old}}$;
- Old solution $x_{\text{old}}$;
- Old slack variables related to $Ax \leq b$, i.e., $\lambda_{\text{old}}$;
- Old slack variables related to $x \geq 0$, i.e., $\mu_{\text{old}}$.

Meanwhile, slack variables $\lambda$ and $\mu$ likewise indicate, for the old solution, which constraints are binding and which are non-binding. Together with the direction of change in the objective $c$, they jointly predict how the old solution will perform in the altered environment.

A potential direction for future research is to develop more comprehensive features, following Assumption 2, to decide when to re-solve. For nonlinear optimization problems, our POC framework remains applicable, together with other carefully designed features tailored to the optimization problem at hand.

### D.3 NETWORK ARCHITECTURES AND HYPERPARAMETERS

After extracting the features via feature engineering, we use them to train both the actor network $\pi_\theta^\xi(\cdot|\cdot)$ and the value network $V_\theta(\cdot)$.

For datasets SC, Mat, SP, FL, GMILP and CA, we employ a binary actor–critic architecture based on a multilayer perceptron. The network consists of three fully connected layers of sizes 512, 256, and 128, each followed by ReLU activations, which are shared between the actor and the critic. The actor network is implemented with a policy head that outputs a single logit, parameterizing a Bernoulli distribution over binary actions, i.e., whether to keep the old solution or to re-solve,

while the critic is implemented with a value head that predicts the state value function. To ensure symmetry at initialization, the weights and bias of the policy head are set to zero, yielding an initial action probability of 0.5.

For SPP and TSP, however, since these two optimization problems have larger scales, we adopt a deeper residual MLP as the shared backbone for the actor network to better capture the structure of the MILPs. Concretely, an input linear layer projects the features to width 1024, followed by three residual blocks, composed of a linear layer, a ReLU activation and another linear layer with a skip connection, and a LayerNorm after each block to stabilize optimization. A bottleneck module, say a ReLU activation, a 256-dimensional linear layer and another ReLU activation, then compresses the representation, upon which we place two linear heads, namely a policy head and a value head as before. This residual, wider architecture improves gradient flow and representation capacity, which we find beneficial for SPP and TSP datasets without changing the training protocol for the critic or the loss definitions.

We use off-the-shelf PPO hyperparameters with no task-specific tuning, and we hypothesize that targeted fine-tuning would further improve our POC framework's performance. We set the discount factor $\gamma = 0.9$, use parameter $\epsilon = 0.2$ for clipping, and set value function coefficient $c_1 = 0.5$ and entropy coefficient $c_2 = 0.01$. We use the `AdamW` optimizer with a learning rate of 0.0001. The policy $\pi_\theta^\xi$ is updated once every 100 epochs, with training capped at 1500 epochs, and we select the model with the lowest cumulative loss on the validation set and evaluate it on the test set. All experiments are conducted on a single NVIDIA H100 GPU. At last, before initiating the POC framework, an initial feasible solution is required. For convenience, we assume the cost vector $c$ to be an all-ones vector in order to obtain this initial solution.

In the ablation study, we investigate the performance of alternative architectures on the task of deciding when to re-solve, such as graph neural networks (GNN), which have proven effective in representing MILP instances (Gasse et al., 2019; Scavuzzo et al., 2022; Li et al., 2024b; Zhang et al., 2024). We further study how model performance varies with hyperparameter settings, for example, how adjusting $\gamma$ can reduce the variance of the value function.

### D.4    BASELINES

In this section, we present the specific implementation details of the baselines.

**ADWIN-5%.**    ADWIN-5% (Bifet & Gavalda, 2007) aims to identify essentially distinct change points with a probability of 5%, and uses them to trigger re-solving. We employ `changeforest` as the change point detector to examine whether the distribution of the objective has shifted. By setting its p-value, we search for change points at each time step with a 5% significance level, and retain the most recent change point to distinguish the current objective distribution. We observe that real-world data are often highly noisy, which leads to fluctuations in the locations of change points identified by the detector. We argue that such fluctuations do not represent essentially distinct changes. For example, given $c_1, ..., c_{100}$, the detector may report the last change point at 75, while for $c_1, ..., c_{101}$, it may return 74. In practice, we do not regard these as different change points, nor should they trigger a re-solve. To address this, we fine-tune a threshold on the validation set that only when the distance between a newly detected change point and the previous one exceeds this threshold do we consider it as identifying a genuinely distinct change point, which then triggers a re-solve.

**CARA-P.**    CARA-P (Mahadevan & Mathioudakis, 2023) periodically triggers re-solving. Since the initial solution lacks observations of the environment, we enforce the algorithm to re-solve the corresponding MILP immediately after the first real observation of the environment. Similarly, we fine-tune the re-solving period on the validation set and apply the optimal period identified there for testing on the test set.

**UPF.**    We follow Florence et al. (2025) to implement UPF. Specifically, we enumerate all possible combinations of the solution time and real time $t$ to construct the training dataset. Note that since $T$ in our setting is on the order of hundreds to thousands, each time series produces roughly $\frac{T^2}{2}$

combinations. Consequently, UPF exhibits high computational complexity and low efficiency in our case.

Similarly, we use `ElasticNet` to train the predictor. For the real-world datasets SPP, TSP, and CA, we standardize the features before regression. We adopt the feature construction in Florence et al. (2025). Specifically, if the previous solution uses $c_{\text{old}}$ to estimate the mean of the objective, then the resulting features are as follows.

- The acquisition time of the old solution;

- Real time $t$;

- Distribution shift in the objective, namely, $\frac{\sum_{j=\iota}^{t-1} c_j}{t - \iota} - c_{\text{old}}$.

Meanwhile, although CARA-T and CARA-CT are not directly applicable to our setup, we draw on their notion of staleness cost and include it as a feature in UPF, leading to a slight improvement in performance.

- Staleness cost $\|\frac{\sum_{j=\iota}^{t-1} c_j}{t - \iota} - c_{\text{old}}\|_2$.

When we decide to re-solve, we cannot observe future $c_t$ in the online setting. Following Florence et al. (2025), we assume that future $c_t$ and the objective at the time of re-solving are drawn from the same distribution. Under this assumption, the staleness cost reduces to zero. Once the UPF predictor is trained, we compare the total optimization loss between re-solving and not re-solving at each decision point. If the difference exceeds the re-solving cost, we perform a re-solve; otherwise, we continue to use the previous solution.

**LBwCP.** For LBwCP, we provide a lower bound on the cumulative loss by ignoring the re-solving cost. Without the re-solving cost, we can re-solve the MILP at every time step. For the synthetic datasets, since the true change points are known, we use these locations to estimate the environment, thereby avoiding the additional loss introduced by the change point detector. LBwCP reflects the minimum achievable optimization loss given the uncertainty of the environment.

**LBwoCP.** Unlike LBwCP, LBwoCP does not have access to the true change point locations. Hence, it reflects how much cumulative loss arises from the non-negligible re-solving cost when using the same change point detector. This provides a decomposition of the cumulative loss, viz.,

$$\text{CL} = \text{LBwCP} + (\text{LBwoCP} - \text{LBwCP}) + (\text{CL} - \text{LBwoCP}).$$

The gap between LBwoCP and LBwCP corresponds to the unavoidable loss introduced by the change point detector, while the difference between an algorithm's cumulative loss and LBwoCP stems from the re-solving cost itself and the resulting reduction in the number of re-solves, which leads to less accurate environment estimation and consequently higher loss.

D.5 RESULTS

In the first experiment, we examine the performance of the POC framework and other baselines across eight datasets. To avoid falling into the trivial cases of the first and third segments of re-solves illustrated in Figure 6, we set 30 seconds as the unit for SPP and TSP. For CA, we set the unit to 10. Experimental results indicate that varying the unit produces similar outcomes. For ADWIN-5% and CARA-P, the corresponding thresholds and periods are reported in Table 2. We find

Table 2: ADWIN-5% threshold and CARA-P period for different datasets.

| Dataset | SC | Mat | SP | FL | GMILP | SPP | TSP | CA |
|---|---|---|---|---|---|---|---|---|
| ADWIN-5% threshold | 1 | 2 | 3 | 5 | 1 | 19 | 28 | 10 |
| CARA-P period | 15 | 11 | 15 | 16 | 15 | 20 | 20 | 24 |

that for real-world datasets, such as the New York GTFS dataset, ADWIN-5% requires a relatively large threshold. This is because real-world environments are highly noisy, making the change point

detector prone to detecting spurious change points. A larger threshold helps filter out essentially redundant change points and prevents excessive re-solving.

In the second experiment, we vary the re-solving cost from 5 to 50. To make the setting more practical, we also use the change point detector to identify change points during training, rather than directly relying on the true change point locations. We find that due to the delay and inaccuracy in change point detection, the cumulative loss increases slightly, for example, when $C = 10$, it rises from 2280.09 to 2375.39. Besides, the total number of re-solves decreases from 18.90 to 16.70, which reflects that as uncertainty increases, the benefit of re-solving diminishes, and the model tends to conservatively rely on the old solution. We first present the experimental results in Table 3.

Table 3: Experimental Results: We report the cumulative loss and the number of re-solving events for distinct re-solving cost $C$. In the tables, the best results are highlighted in **bold**, and the second-best results are underlined.

| Algo | $C = 5$ | | $C = 10$ | | $C = 20$ | |
|---|---|---|---|---|---|---|
| | CL ($\downarrow$) | # R-S | CL ($\downarrow$) | # R-S | CL ($\downarrow$) | # R-S |
| ADWIN-5% | 2377.04 | 74.10 | 2747.42 | 74.10 | 3421.23 | 59.70 |
| CARA-P | 2792.29 | 91.00 | 3243.08 | 67.00 | 4012.23 | 39.00 |
| UPF | 5301.43 | 688.60 | 7030.22 | 517.00 | 8988.62 | 356.10 |
| POC (ours) | **2341.01** | 31.70 | **2375.39** | 16.70 | **2670.83** | 12.40 |
| LBwoCP | 1859.58 | - | 1859.58 | - | 1859.58 | - |
| Algo | $C = 30$ | | $C = 40$ | | $C = 50$ | |
| | CL ($\downarrow$) | # R-S | CL ($\downarrow$) | # R-S | CL ($\downarrow$) | # R-S |
| ADWIN-5% | 4018.23 | 59.70 | 4797.57 | 58.00 | 5377.57 | 58.00 |
| CARA-P | 4402.23 | 39.00 | 5243.54 | 27.00 | 5513.54 | 27.00 |
| UPF | 10184.67 | 277.40 | 11061.07 | 229.90 | 11750.65 | 197.70 |
| POC (ours) | **2858.57** | 11.40 | **2982.99** | 10.30 | **3331.65** | 9.80 |
| LBwoCP | 1859.58 | - | 1859.58 | - | 1859.58 | - |

Subsequently, in Table 4, we report the thresholds of ADWIN-5% and the periods of CARA-P for different re-solving costs $C$. We observe that as the re-solving cost increases, the POC framework as well as all other baselines tend to reduce the number of re-solves in order to balance the optimization loss and the re-solving cost.

Table 4: ADWIN-5% threshold and CARA-P period for distinct re-solving costs.

| Re-solving Cost | $C = 5$ | $C = 10$ | $C = 20$ | $C = 30$ | $C = 40$ | $C = 50$ |
|---|---|---|---|---|---|---|
| ADWIN-5% threshold | 1 | 1 | 8 | 8 | 9 | 9 |
| CARA-P period | 11 | 15 | 26 | 26 | 37 | 37 |

Finally, we investigate the robustness of our POC framework, namely, how the performance of the learned policy $\pi_\theta$ is affected when it is trained with an incorrect re-solving cost. We present our results in Table 5. We find that our POC framework exhibits strong robustness. Even when the re-solving cost is mis-specified by a factor of ten, the performance of POC decreases by less than 25%, and on average the degradation is below 5%. We observe that in some cases a mis-specified $C$

Table 5: Cumulative loss of the policy trained with a mis-specified re-solving cost $C'$, evaluated under the true re-solving cost $C$.

| Real $C$ vs. Provided $C'$ | $C = 5$ | $C = 10$ | $C = 20$ | $C = 30$ | $C = 40$ | $C = 50$ |
|---|---|---|---|---|---|---|
| $C' = 5$ | 2341.01 | 2499.51 | 2816.51 | 3133.51 | 3450.51 | 3767.51 |
| $C' = 10$ | 2291.89 | 2375.39 | 2542.39 | 2709.39 | 2876.39 | 3043.39 |
| $C' = 20$ | 2484.83 | 2546.83 | 2670.83 | 2794.83 | 2918.83 | 3042.83 |
| $C' = 30$ | 2573.57 | 2630.57 | 2744.57 | 2858.57 | 2972.57 | 3086.57 |
| $C' = 40$ | 2622.49 | 2673.99 | 2776.99 | 2879.99 | 2982.99 | 3085.99 |
| $C' = 50$ | 2890.65 | 2939.65 | 3037.65 | 3135.65 | 3233.65 | 3331.65 |

even leads to better performance. This may stem from the interaction between the discount factor $\gamma$ and re-solving cost $C$. We use the GMILP dataset since it represents the most general MILP dataset. Investigating the robustness of the POC framework on other real-world datasets and developing automated hyperparameter tuning are promising directions for future research.

### D.6 DESIGN CHOICES ANALYSIS AND ABLATION STUDY

We first investigate using linear programming (LP) as a warm start to decide whether to re-solve the MILP. We evaluate this approach on the GMILP dataset, and within our problem scale, solving the LP is 25 times faster than solving the MILP. We first relax the integrality constraints to examine the performance of our POC framework on LP in Table 6. We find that our framework also outperforms other baselines in the LP setting, indicating its compatibility with a broader range of optimization problems.

Table 6: Experimental Results: We report the cumulative loss, the number of re-solving events and fine-tuned hyperparameters for all algorithms on LP problems. In the tables, the best results are highlighted in **bold**, and the second-best results are underlined.

| Algo | LP | | | |
|---|---|---|---|---|
| | CL ($\downarrow$) | # R-S | ADWIN-5% threshold | CARA-P period |
| ADWIN-5% | 2741.25 | 74.10 | 1 | - |
| CARA-P | 3242.55 | 67.00 | - | 15 |
| UPF | 7072.21 | 521.20 | - | - |
| POC (ours) | **2278.81** | 24.50 | - | - |
| LBwCP | 1632.81 | - | - | - |
| LBwoCP | 1861.28 | - | - | - |

Subsequently, we fine-tune the actor network obtained from LP as a warm start, incorporating integrality constraints to decide when to re-solve the corresponding MILP problem. The results are reported in Table 7. We observe that the number of epochs required for convergence decreases significantly from 600 to 300, while the cumulative loss increases slightly. This suggests that the LP warm start may sacrifice certain observations specific to MILP, providing a case study of the trade-off between training efficiency and performance in real-world applications.

Table 7: Performance of POC under cold start and warm start.

| Start Scheme | GMILP | | |
|---|---|---|---|
| | CL ($\downarrow$) | # R-S | Epochs |
| Cold Start | 2280.09 | 18.90 | 600 |
| Warm Start | 2336.60 | 19.40 | 300 |

Next, we investigate the impact of different discount factors $\gamma$ on model performance. Since we are dealing with high-frequency data flows, the probability of environmental changes between consecutive time steps is small, suggesting that the theoretically optimal $\gamma$ should be close to 1. However, increasing $\gamma$ also amplifies the variance of the value function, which can lead to gradient explosion and training collapse. A moderately sized $\gamma$ can thus effectively mitigate these risks. We consider $\gamma = 0.85, 0.90$, and $0.95$, and the results in Table 8 show that $\gamma = 0.90$ provides a good balance between theoretical guidance and empirical training stability.

Table 8: Performance of POC under different discount factors.

| Discount Factor | GMILP | |
|---|---|---|
| | CL ($\downarrow$) | # R-S |
| $\gamma = 0.85$ | 2377.30 | 32.10 |
| $\gamma = 0.90$ | 2280.09 | 18.90 |
| $\gamma = 0.95$ | 2292.15 | 21.40 |

Beyond MLPs, GNNs are another widely used architecture capable of representing the structural information of MILPs. Besides the features discussed above, we additionally introduce features that

are standard in prior GNN work (Gasse et al., 2019; Paulus et al., 2022), as follows. We denote the previous LP approximation solution as $x_{\text{old}}^{LP}$.

- Normalized objective coefficient $\frac{\sum_{j=\iota}^{t-1} c_j}{t-\iota} / \| \frac{\sum_{j=\iota}^{t-1} c_j}{t-\iota} \|_2$;

- Existing LP solution value $x_{\text{old}}^{LP}$;

- Existing LP solution value fractionality $x_{\text{old}}^{LP} - \lfloor x_{\text{old}}^{LP} \rfloor$;

- Unshifted side normalized by row norm, $b_i / \|A_i\|_2$ for all $i$;

- Cosine similarity of the row with the objective, $\cos(\frac{\sum_{j=\iota}^{t-1} c_j}{t-\iota}, A_i)$ for all $i$;

- Row value equals right-hand side, $\mathbb{1}\{A_i x_{\text{old}}^{LP} = b_i\}$ for all $i$;

For all denominators, we add $10^{-8}$ to avoid division by zero. At the same time, we construct the edge index from the nonzero elements of $A$, which allows the GNN to recover the structural information of the MILP.

We use a GNN architecture consisting of two input encoders, a sequence of graph convolutional layers, and separate heads for the actor network and value function. Specifically, variable and constraint features are first projected into a shared latent space of size 128 through independent linear transformations. These embeddings are concatenated and then passed through two graph convolutional network layers, each with hidden dimension 128, which propagate information along the bipartite graph defined by the variable–constraint edge index, thereby capturing the structural dependencies of the underlying MILP instance. After message passing, variable representations are extracted and aggregated by mean pooling within each graph to obtain a compact graph-level embedding. This embedding is then fed into two separate output modules, the policy head and the value head, implemented as a small feedforward network with a hidden layer of width 128 and ReLU activation, to predict state values.

We conduct comparative experiments on both a synthetic dataset (GMILP) and a real-world dataset (SPP). As shown in Table 9, we find that the performance of GNNs is rather poor, which is due to

Table 9: Performance of POC under different network architectures.

| Architecture | MILP | | SPP | |
|---|---|---|---|---|
| | CL ($\downarrow$) | # R-S | CL ($\downarrow$) | # R-S |
| MLP | 2280.09 | 18.90 | 825.96 | 15.50 |
| GNN | 4245.93 | 105.00 | 1151.07 | 11.70 |

the structural characteristics of our problem. In the GMILP dataset, the constraint matrix $A$ is often dense, which leads to an excessive number of nonzero edges in the edge index. Specifically, the dimensionality of the edge index is 20000, while all other features together have a dimensionality of only 803. As a result, the GNN does not effectively exploit the structure of the MILP and instead tends to overfit, producing large uncertainty. This leads to unsatisfactory generalization performance on the test set. On the SPP dataset, we observe the same issue. In this case, 2504 dimensions are devoted solely to representing the structure of the matrix $A$, whereas the features directly relevant to predicting when to re-solve amount to only 2257 dimensions. Compared with the GMILP dataset, this alleviates the severity of the imbalance, yet the mismatch in dimensionality still induces overfitting and high variance. Consequently, while using a GNN to predict when to re-solve on the SPP dataset does not perform as poorly as in the GMILP case, it still underperforms relative to an MLP.

These results suggest that GNNs are better suited for sparse MILPs, where the structural representation remains manageable. In contrast, applying GNNs to dense MILPs yields diminishing returns, as the overwhelming number of edges not only hampers learning efficiency but also increases cumulative loss.

In real-world settings, the distribution of the objective may also undergo continuous small changes, making the use of segment averages between change points no longer optimal. To investigate this, we employ the SPP dataset and study two other commonly used approaches, say exponential moving averages (EMA) and windowing. For EMA, we set the smoothing factor to 0.9. For windowing, we treat every 20 consecutive time steps as a segment, then wait for 300 time steps before concatenating

another 20-step segment. By repeating this process, we construct sequences of 300 time steps. We assume that segments separated by 300 time steps originate from different distributions, and we regard the concatenation points as change points. The results are summarized in Table 10. We

Table 10: Performance of POC under different weighting schemes.

| Weighting Scheme | SPP | |
|---|---|---|
| | CL ($\downarrow$) | # R-S |
| CPD | 825.96 | 15.50 |
| EMA | 751.86 | 11.80 |
| Windowing | 1138.01 | 8.00 |

observe that on real-world datasets, EMA achieves a lower cumulative loss compared to our change point detector. Although EMA lacks theoretical guarantees, it is well-suited for real-time operations, as real environments typically evolve continuously at every moment. This provides a practical recipe for deploying our POC framework in real-world settings.

Finally, we conduct an ablation study to examine whether the number of observations is a beneficial feature that can enhance model performance. As shown in Table 11, incorporating the number of observations, both from the previous solution and from the current re-solve, reduces the cumulative loss by about 15%. This supports our approach of categorizing re-solves into two types, namely, those triggered by environmental changes, and those enabled by having more observations to better estimate the environment. It also indicates that including the number of observations as a feature helps improve decision-making for the latter type of re-solve.

Table 11: Performance of POC under different features.

| Feature Engineering | GMILP | |
|---|---|---|
| | CL ($\downarrow$) | # R-S |
| Including Sample Size | 2280.09 | 18.90 |
| Excluding Sample Size | 2676.41 | 17.10 |

This concludes the appendix. We only use LLMs for simple writing checks, such as grammar errors. Since LLMs do not play a significant role, they should not be regarded as a contributor.

