# OpenReview forum: "Solve Smart, Not Often: Policy Learning for Costly MILP Re-solving"
_ICLR.cc/2026/Conference — ICLR 2026 Conference Withdrawn Submission_

### Official Review · Reviewer_5JPi · 2025-10-24

**Soundness:** 3
**Presentation:** 1
**Contribution:** 2
**Rating:** 4
**Confidence:** 3

**Summary:**

This paper addresses the important real-world problem of determining when to re-solve computationally expensive Mixed-Integer Linear Programs (MILPs) in dynamic environments. The authors propose a framework called Proximal Policy Optimization with Change Point Detection (POC) that balances solution quality and solving cost. The paper establishes theoretical relationships between the number of re-solves and re-solving cost, and demonstrates through extensive experiments on eight synthetic and real-world datasets that POC consistently outperforms existing baselines by 2%-17%. The work fills a significant gap in the literature by introducing real-time MILP benchmarks and evaluation criteria.

**Strengths:**

1. Addresses a practically important problem with real-world relevance in operations research and optimization.
2. Provides both theoretical analysis and extensive experimental validation across diverse datasets.
3. The theoretical results (Theorems 1 and 2) provide meaningful insights into the relationship between re-solving frequency and optimization loss.

**Weaknesses:**

1. The paper doesn't sufficiently discuss the computational overhead of the change point detection component, which might be significant for real-time applications.
2. The paper claims to address "large time horizon and non-MDP structure" challenges, but doesn't clearly explain how POC overcomes these limitations compared to other reinforcement learning approaches.
3. The experimental section could benefit from more detailed analysis of the trade-offs between solution quality and computational cost.
4. The presentation of Figure 2 is somewhat oversimplified, while the layout of Figures 3-5 is suboptimal.

**Questions:**

1. The paper fails to mention which solver was used, and it is unclear if tests were run on multiple solvers.
2. The description of the dataset is lacking; the number of variables and constraints for the MILP problems should be included.
3. Could you elaborate on how your framework specifically handles the non-MDP structure of MILP re-solving problems, and why vanilla RL methods fail in this context?
4. The paper mentions "real-time MILP benchmarks" - could you provide more details about these benchmarks and how they differ from existing MILP benchmarks?

---

### Official Review · Reviewer_cybh · 2025-10-29

**Soundness:** 3
**Presentation:** 2
**Contribution:** 2
**Rating:** 2
**Confidence:** 2

**Summary:**

This paper addresses the practical challenge of determining when to re-solve MILPs in dynamic environments with high-frequency data streams. The authors propose POC, a framework combining change point detection, feature engineering, and PPO to balance optimization loss and re-solving costs. Theoretical analysis characterizes the relationship between environment changes and discount factors, deriving bounds on re-solving frequency. Experiments show that POC reduces cumulative loss by 2%–17% compared to baselines while significantly cutting re-solving events.

**Strengths:**

1. The paper systematically formalizes the understudied problem of cost-aware MILP re-solving in non-stationary environments, highlighting real-world constraints
2.  Theoretical results are provided that link environment change probability to discount factors in RL and prove structural properties of re-solving intervals.

**Weaknesses:**

1. The dynamic enviroment is tackled via check point dectection, which is not contributed by the authors. The settings of both dynamic environment and unobservable $c_t$ seem mixed together and hinder the insights a little bit.
2. When constraints are static, **prior solutions may accelerate subsequent re-solving** (e.g., via cutting planes), such accumulative advantage should not be neglected.
3. **Notations**: $c_t^Tx$ should be $c_t^\top x$ (\top), since $T$ is the given timeframe. Moreover, I fail to find the definition of $t_k^*$ in Theorem 2.

**Questions:**

1.  Intuitively the re-solving cost is hard to exactly quantify, but is crucial to $CL(\pi)$. How should practitioners set the re-solving cost $C$ in real applications?
2.  An easier setting is that $c_t$ is varying but can be observed at time $t$. Do there exist approaches for such setting? If no, why don't the authors consider the more fundamental case first; if yes, then is POC applicable to such case and what is POC's advantage?

---

### Official Review · Reviewer_YKwD · 2025-10-30

**Soundness:** 3
**Presentation:** 2
**Contribution:** 2
**Rating:** 4
**Confidence:** 2

**Summary:**

This paper addresses the problem of determining when to re-solve a Mixed-Integer Linear Program (MILP) in dynamically changing environments while accounting for the cost that arises from repeated re-solving. To tackle this problem, the authors propose the POC framework, which combines Proximal Policy Optimization (PPO) with Change-Point Detection (CPD). The paper also provides theoretical analysis on structural properties such as an upper bound on the re-solving frequency and increasing intervals between re-solves under stable environments. The authors construct eight MILP-based benchmarks to evaluate their method and empirically demonstrate that the proposed approach reduces cumulative optimization loss more effectively than existing methods.

**Strengths:**

- This paper proposes a method to determine when to re-solve MILP problems while considering re-solving costs. Although this topic has not been extensively explored in prior work, it is practically relevant for certain industrial applications.

- The paper performs theoretical analysis on structural properties, including an upper bound on re-solving frequency and increasing intervals when the environment remains stable.

- To evaluate the proposed approach, the authors construct eight benchmarks, establishing a foundation for future research in the field.

- Based on the provided datasets, the authors empirically demonstrate reductions in cumulative loss compared to existing methods.

**Weaknesses:**

- This work assumes a single constant re-solving cost; however, in real-world scenarios, quantifying such costs can be challenging for various reasons. As a result, applying the proposed approach directly to practical systems may be limited.

- POC requires additional computation for determining re-solving times, yet the paper does not sufficiently discuss how this overhead affects the overall cost structure.

- The proposed POC framework introduces additional complexity compared to existing approaches, requiring further development and maintenance efforts to integrate into operational environments. In industrial settings, re-solving decisions may not always be critical enough to justify adopting a more complex system.

**Questions:**

None

---

### Official Review · Reviewer_zToV · 2025-11-01

**Soundness:** 1
**Presentation:** 1
**Contribution:** 2
**Rating:** 2
**Confidence:** 3

**Summary:**

This paper proposes Proximal Policy Optimization with Change Point Detection (POC) that learns when to re-solve a large-scale Mixed-Integer Linear Programming (MILP) problem by using reinforcement learning (RL). POC is design to balance the accuracy and the computation cost in a situation where the volume and distribution of input data is dynamic. This paper evaluates POC on eight benchmark datasets.

**Strengths:**

- S1. [Motivation] This paper proposes a RL-based change point detection method for solving a large-scale MILP.

**Weaknesses:**

- W1. [Formulation] The problem formulation is rather unclear. It seems like that POC aims to lean when to re-solve a large-scale MILP problem by using RL. However, this paper does not provide a problem formulation based on Markov Decision Process (MDP). If this paper formulates the problem as a constraint RL, this paper needs to provide a proper formulation.

- W2. [Presentation] The presentation of this paper needs to be revised. The section structure of this paper does not seem effective. For example, Section 2 (i.e., Preliminaries) and 3 (i.e., Theoretical Analysis) are relatively long (about three pages), compared to Section 4 (i.e., Method). On the other hand, the related works section (i.e., Section 1.1) is very short (only one paragraph).

- W3. [Relevance] I am not sure whether the problem addressed in this paper is relevant with topics of interest to the ICLR research community.

**Questions:**

- Q1. What is the state space, the action space, the transition function, the reward function, and discount factor of the policy?

- Q2. Do the authors measure the amount of computational cost induced by POC?

---

### Note · Authors · 2025-11-19

I have read and agree with the venue's withdrawal policy on behalf of myself and my co-authors.